# Predictors of Impaired Reperfusion in ST-Elevation Myocardial Infarction Treated with Primary PCI: Preliminary Results from COMA.NET Project

**DOI:** 10.3390/diagnostics16010149

**Published:** 2026-01-02

**Authors:** Maciej Południewski, Emil Julian Dąbrowski, Piotr Pogorzelski, Michał Łuczaj, Julia Kobylińska, Joanna Kruszyńska, Marcin Kożuch, Sławomir Dobrzycki

**Affiliations:** Department of Invasive Cardiology, Medical University of Białystok, Kilińskiego 1, 15-089 Białystok, Poland

**Keywords:** impaired reperfusion, no reflow, STEMI

## Abstract

**Background:** The no-reflow phenomenon remains a frequent and clinically significant complication in patients with ST-segment elevation myocardial infarction (STEMI) despite advances in primary percutaneous coronary intervention (pPCI). Its determinants are multifactorial and not fully established. This study aimed to identify independent predictors of impaired reperfusion after pPCI. **Methods:** In this prospective study, 100 consecutive STEMI patients treated with successful pPCI in a high-volume tertiary center were analyzed. Impaired reperfusion was defined as ST-segment resolution < 50% or final TIMI flow < 3. Clinical characteristics, laboratory findings, including platelet reactivity, and detailed angiographic and procedural parameters were collected. Independent predictors were evaluated using multivariable logistic regression. Thirty-day and twelve-month mortality were assessed with Kaplan–Meier analysis. **Results:** Impaired reperfusion occurred in 39% of patients. Compared with the normal reperfusion group, patients with noreflow were older, had lower left ventricular ejection fraction, eGFR, longer ischemia times, and more often presented with anterior STEMI. Platelet reactivity did not differ between groups. Four variables independently predicted impaired reperfusion: longer pain-to-balloon time (OR 1.05 per 10 min, 95% CI 1.02–1.07; *p* < 0.001), anterior myocardial infarction (OR 5.05, 95% CI 1.14–22.38; *p* = 0.03), use of predilatation (OR 7.66, 95% CI 1.78–32.9; *p* = 0.006), and higher Killip–Kimball class (OR 7.69, 95% CI 1.88–31.38; *p* = 0.004). Impaired reperfusion was associated with significantly higher mortality at 30 days (1.6% vs. 10%; *p* < 0.001) and 12 months (3.2% vs. 25.6%; *p* < 0.001). **Conclusions:** In this prospective STEMI cohort, impaired reperfusion was frequent and strongly associated with adverse short- and long-term outcomes. Ischemia duration, infarct location, hemodynamic status, and procedural strategy were key determinants of noreflow, while platelet reactivity showed no significant association.

## 1. Introduction

Despite noticeable improvement in the management of ST segment elevation myocardial infarction (STEMI), insufficient microcirculatory perfusion, often described as the no-reflow phenomenon, still reduces the therapeutic efficacy [1]. The preferred treatment strategy for STEMI is an early reperfusion by opening the occluded epicardial coronary artery with primary percutaneous coronary intervention (pPCI) [2]. Even after timely and technically successful restoration of blood flow in an infarct-related artery, some patients remain at increased risk of adverse clinical outcomes, potentially related to impaired myocardial reperfusion [3]. The no-reflow phenomenon (NRP) is a multifactorial process, most accurately assessed by invasive thermodilution or Doppler guidewire measurements of coronary flow reserve (CFR) or noninvasive cardiac magnetic resonance. However, in daily clinical practice, more accessible methods such as angiographic scales (TIMI, TMPG, MBG, and cTFC) and ST-segment resolution (STR) are used to assess NRP [3,4]. Its incidence widely varies between 10 and 50% of STEMI patients, depending on the used definition and assessment methods [1]. Among underlying factors of impaired reperfusion, we can distinguish myocardial ischemia, distal embolization, reperfusion injury, and individual susceptibility [1].

It is well established that NRP is associated with worse clinical outcomes, causing adverse left ventricle remodeling, cardiac arrhythmias, and heart failure, and is an independent predictor of death [4]. Those various adverse outcomes and lack of one effective target-focused treatment make it crucial to search for risk factors, which can help identify high-risk patients and ultimately improve the prognosis [5]. Some studies have suggested that no reflow might be predicted in patients presenting prolonged ischemia, advanced age, diabetes, or renal impairment [6,7,8]. However, several studies highlight the important role of platelets. Platelets might contribute to the no-reflow phenomenon in acute myocardial infarction, largely through microembolization of atherosclerotic debris, thrombi, and platelet aggregates following reperfusion therapies such as thrombolysis, angioplasty, or stenting [9,10]. Nevertheless platelet reactivity does not universally appear as a reproducible predictor in the literature. Only a few studies focus directly on platelet impact on impaired reperfusion. Kuliczkowski et al. investigated a small group of diabetic patients with no reflow and found that higher platelet reactivity was associated with an increased risk of impaired reperfusion [11]. Huczek et al. examined a broader population of STEMI patients undergoing primary percutaneous coronary intervention and reported that platelet reactivity is an independent predictor of myocardial reperfusion, although the findings should be interpreted with caution due to the study’s limitations [12]. Notably, several studies investigating the no-reflow phenomenon have not included platelet reactivity among the variables analyzed [13,14].

This indicates that the risk factors for the no-reflow phenomenon remain incompletely recognized, leaving room for further research. To address this gap in evidence, we conducted a comprehensive prospective evaluation of predictors of impaired reperfusion following pPCI in a high-volume tertiary cardiac center in Poland.

## 2. Materials and Methods

### 2.1. Study Population

A prospective analysis was conducted on patients admitted urgently to the Department of Invasive Cardiology, Medical University of Białystok, from August 2015 to August 2017 with a diagnosis of STEMI. The analysis included all consecutively admitted patients who met the STEMI criteria, signed Informed Consent Form (ICF), and underwent successful primary angioplasty of the infarct-related artery (IRA). Inclusion criteria were age over 18 years, chest pain with a duration >20 min. and <24 h, ST segment elevation fulfilling STEMI criteria according to the IV definition of MI, successful primary angioplasty of IRA, use of a loading dose of clopidogrel, and written informed consent for participation in the study. The exclusion criteria were cardiogenic shock, significant residual stenosis or IRA after PCI, intraventricular conduction abnormalities that complicate ST-segment assessment, cardiac pacing, digoxin use, chronic dual antiplatelet therapy before inclusion, GP IIb/IIIa administration prior blood sample collection, and prasugrel or ticagrelor administration.

### 2.2. Definitions and Outcome

Electrocardiographic recordings were evaluated at admission and approximately 60 min after pPCI. The ST segment elevation resolution was assessed in the lead with initially the highest ST elevation. Two independent, experienced interventional cardiologists assessed the coronary angiograms using the TIMI, MBG, TMPG, and cTFC scales. Additionally, the severity of atherosclerotic lesions was assessed using the SYNTAX score I. The course of the procedure was analyzed with particular attention to the techniques and devices used.

At the time of admission to the catheterization laboratory, arterial blood was drawn from the vascular sheath at the beginning of the procedure and venous blood on the 5th day of hospitalization. Blood was collected into hirudin-containing tubes for the assessment of platelet reactivity induced by adenosine diphosphate (ADP) and arachidonic acid (AA). The analysis was performed using the Multiplate Analyzer (Roche, Switzerland) with ASPItest (assessment of platelet aggregation inhibition by acetylsalicylic acid) and ADPtest (assessment of platelet aggregation inhibition by P2Y12 receptor inhibitors). Standard biochemical parameters, echocardiographic findings, and clinical assessment were also subjected to detailed analysis.

Based on the available literature and to ensure an accurate assessment of the complex mechanisms underlying impaired tissue reperfusion, we used a combination of angiographic and electrocardiographic scales. Among the angiographic parameters, a TIMI flow grade < 3 demonstrated the strongest predictive value for one-year mortality, whereas, among the electrocardiographic indicators, an ST-segment resolution (STR) < 50% showed the best prognostic accuracy. The integration of these two assessment methods (TIMI < 3 and/or STR < 50%) provided the most comprehensive characterization of the multifactorial mechanisms underlying the no-reflow phenomenon and demonstrated a strong association with one-year mortality (OR 3.2; 95% CI: 1.45–7.00; *p* = 0.004).

### 2.3. Statistical Analyses

Continuous data with a normal distribution are reported as means with standard deviations (SD) or medians with interquartile ranges (IQR) in cases of non-normal distribution. Categorical data are presented as absolute counts (N) and percentages (%). Continuous variables were compared using either Student’s *t*-test or the Wilcoxon rank-sum test, depending on the data distribution and categorical variables analyzed using the chi-square test.

Predictors of impaired reperfusion were assessed using logistic regression. Firstly, potential baseline and angiographical variables were included in univariate analysis as independent variables (Appendix A). Then, all variables with *p*-value less than 0.2 were included in a multivariate model. After collinearity analysis, variables with variance inflation factor (VIF) > 5 or tolerance < 0.2 were excluded from the initial model. Finally, the model’s performance was assessed using an AUC ROC and Hosmer–Lemeshow test, which showed good discrimination (ROC AUC 0.909) and calibration (Hosmer–Lemeshow *p* = 0.79). The final list of variables included in multivariate model, collinearity, ROC curve, and Hosmer–Lemeshow analyses are presented in the Section 3, Appendix A, and Appendix A.

Mortality rates were calculated using log-rank test at 30 and 365 days after pPCI. Kaplan–Meier curves illustrated time-to-event and absolute event rates across exposure groups.

For all analyses, the level of statistical significance was set at *p* < 0.05. Missing values for baseline variables were not imputed using any methods.

All statistical analysis was performed using StataNow/SE version 18.5 for Mac (StataCorp. 2023. Stata Statistical Software: Release 18. College Station, TX, USA: StataCorp LLC.).

### 2.4. Ethical Considerations

This study was approved by the Bioethics Committee of the Medical University of Białystok, Poland (approval no. R-I-002/60/2008 and R-I-002/60A/2013/2013,19 December 2013), and adheres to Helsinki Declaration as revised in 2013.

## 3. Results

### 3.1. Baseline Characteristics

The study group (*n* = 100) was divided into two subgroups depending on the occurrence of impaired reperfusion after successful primary percutaneous coronary intervention (pPCI), defined in this study as the absence of 50% ST segment elevation resolution or TIMI < 3 end-flow in the revascularized artery. Based on the above criteria, NRP was found in 39 (39%) of the patients. Patients with impaired reperfusion were significantly older compared to the group with normal reperfusion (70.6 ± 12.0 vs. 62.8 ± 11.6 years, *p* = 0.002). In addition, this group of patients was characterized by a lower estimated glomerular filtration rate (eGFR 71 (51–92) vs. 86 (72–114) mL/min, *p* = 0.005) and a statistically significantly lower left ventricular ejection fraction (EF 34.3 ± 8.6% vs. 41.3 ± 7.9%, *p* < 0.001). No significant differences in rates of comorbidities such as hypertension, diabetes, hyperlipidemia, atrial fibrillation, and a history of CAD, coronary interventions, or myocardial infarctions on the occurrence of NRP was demonstrated. Notably, in heavy smokers, normal reperfusion in the infarct-related vessel was significantly more frequent after pPCI (57.4% vs. 23.1%, *p* < 0.001).

Duration of pain (466 vs. 163 min, *p* < 0.001), time from the onset of the first symptoms to the implementation of prehospital treatment in the form of antiplatelet drugs (360 vs. 90 min, *p* < 0.001) and time to receive targeted invasive treatment (505 vs. 185 min, *p* < 0.001) was significantly longer in patients with impaired reperfusion compared to the control group.

It was observed that NRP occurred at a higher rate in patients with anterior myocardial infarction (74.4% vs. 41%, *p* = 0.004) as opposed to the group of patients with inferior myocardial infarction, where normal reperfusion was predominant.

There were no significant differences in platelet reactivity between the two groups at admission or on day 5 of hospitalization, even after stratification by pain-to-balloon time intervals (Appendix A). 

At discharge, patients with impaired reperfusion were significantly less likely to be prescribed ACE inhibitors and more likely to receive diuretic therapy.

No significant differences between groups were observed in the prescription rates of antiplatelet agents, β-blockers, statins, MRAs, or proton pump inhibitors. Detailed baseline characteristics are shown in Table 1.

### 3.2. Procedural Data

Patients with impaired reperfusion had higher SYNTAX score (21 ± 9 vs. 17 ± 8, *p* = 0.04) and more often underwent predilatation (87.2% vs. 50%, *p* < 0.001). Also, time of procedure and exposition to radiation were significantly higher in NRP group (40 vs. 32 min, *p* = 0.07; 10.3 vs. 7.6 min, *p* = 0.02). There were no significant differences in use of thrombectomy, number, diameter, and length of stents, nor the volume of used contrast. Detailed procedural data is provided in Table 2.

### 3.3. Follow-Up

Mortality during hospitalization was numerically higher in patients with abnormal reperfusion compared to the control group (*n* = 4 (10.3%) vs. *n* = 1 (1.6%), *p* = 0.054). A substantial increase in mortality associated with NRP was observed within the first 30 days after intervention (log-rank *p* < 0.001) and was sustained in one year observation (log-rank *p* < 0.001). Figure 1 shows Kaplan–Meier curves in both observation periods.

### 3.4. Predictors of Impaired Reperfusion

In multivariate logistic regression (Table 3.), four variables were associated to the increased risk of impaired reperfusion after pPCI. Longer time from pain onset to balloon inflation (pain to balloon time) was associated with a significant increase in the probability of no reflow (OR = 1.05 for every 10 min of delay; *p* < 0.001). Anterior wall MI increased the risk of no-reflow phenomenon (OR = 5.05; 95% CI: 1.14–22.38; *p* = 0.03) while the need to perform predilation was associated with seven-fold higher risk of impaired reperfusion (OR = 7.66; 95% CI: 1.78–32.9; *p* = 0.006). Increase in Killip–Kimball class significantly influenced risk of impaired reperfusion (OR = 7.69; 95% CI: 1.88–31.38; *p* = 0.004). Factors such as smoking, atrial fibrillation, or SYNTAX score I had no prognostic significance. Graphical representation of time-to-balloon impact on mortality is provided in Figure 2.

**Figure 1 diagnostics-16-00149-f001:**
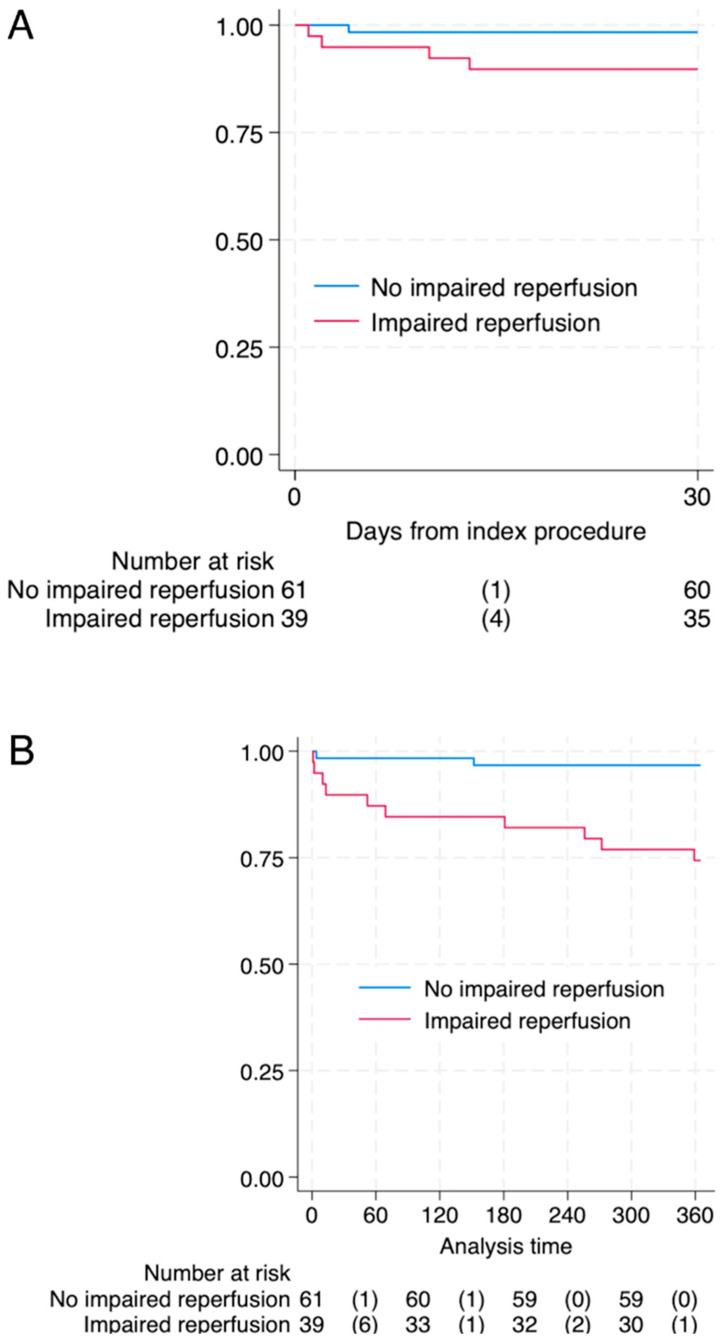
Survival after 30-panel (**A**) and 365 days-panel (**B**) after pPCI in patients with impaired- and non-impaired reperfusion. Log-rank *p*-value <0.001 for both timeframes. Abbreviations: numbers of events (deaths).

**Figure 2 diagnostics-16-00149-f002:**
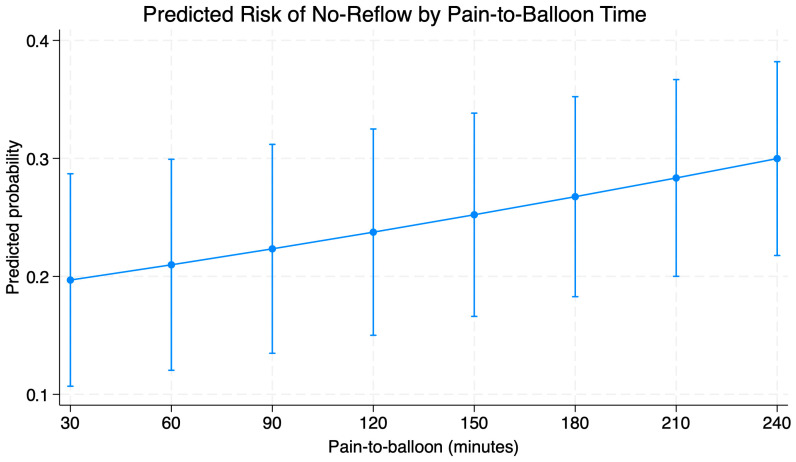
Predicted probability of noreflow based on pain-to-balloon time based on outcomes of multivariable regression.

## 4. Discussion

No-reflow phenomenon is a significant clinical problem in the treatment of STEMI exerting a meaningful adverse effect on patient prognosis. The prevalence of this phenomenon varies between studies, depending on the methodology and definition of impaired reperfusion. In a CathPCI Registry, angiographically diagnosed no reflow (TIMI flow grade < 3) occurred in 2.3% of patients with acute myocardial infarction (6553 out of 291,380) [15]. In a TOTAL study where the methodology for diagnosing impaired reperfusion was similar, this phenomenon occurred in 196 of 1800 patients (10.9%) [16]. In our study impaired reperfusion was defined as either post-procedural ST-segment elevation resolution of less than 50% or a final TIMI flow grade < 3. Such a definition better reflects the complex and multifactorial nature of coronary microcirculatory dysfunction. Applying these criteria, NRP was identified in 39 of 100 patients (39%).

The study population consisted of 100 consecutive patients with myocardial infarction with persistent ST segment elevation who met the inclusion criteria for the study. The clinical characteristics were typical for a population with a high cardiovascular risk. The majority of the patients were men, and a high percentage of hypertension, hyperlipidemia, diabetes, chronic kidney disease, and tobacco abuse were observed. Since impaired reperfusion after percutaneous coronary intervention is a risk factor for poor short- and long-term prognosis, adversely affects left ventricular remodeling, and increases the risk of cardiac arrhythmias, it is reasonable to actively search for risk factors predisposing to this phenomenon. A considerably higher incidence of NRP after successful revascularization was observed in older patients. Moreover, patients with NRP had lower left ventricular ejection fraction, which may be related to impaired reperfusion after successful angioplasty and subsequent myocardial damage. Some studies have found that the presence of certain comorbidities, such as diabetes or renal impairment, may predict the occurrence of NRP [6,7,8]. In our study impaired reperfusion correlated only with lower eGFR, which may indicate the detrimental impact of chronic kidney disease, while hypertension, diabetes, and hyperlipidemia had no significant effect. It is interesting to note that NRP was significantly less common in the group of patients who were chronic smokers. This phenomenon can be explained by the so-called “smokers paradox”, which assumes that smokers who have had a myocardial infraction have a better prognosis due to the younger age at which acute coronary syndrome occurs and the presence of less advanced atherosclerotic changes in these patients [17,18].

Microembolization is one of the main mechanisms of tissue reperfusion disorders after successful coronary angioplasty in patients with STEMI. It causes mechanical arrest of blood flow in the microcirculation, and biologically active embolic material activates the local immune response, causing the release of inflammatory mediators, which leads to capillary constriction [19,20]. Previous studies have indicated that enhanced inhibition of coagulation may decrease the risk of microcirculatory impairment after successful restoration of epicardial blood flow, whereas high platelet reactivity during antiplatelet therapy can significantly compromise tissue reperfusion. [12,21,22,23].

In our study, we examined platelet activity upon admission and on the fifth day of hospitalization, which showed no significant differences between patients with impaired reperfusion and the control group. The discrepancy between our findings and those of previous studies may be attributed to relatively long pre-hospital delays observed in our cohort. As has been demonstrated in the literature, pathophysiological mechanisms underlying impaired reperfusion are time-dependent. During the early phase of myocardial infarction, impaired tissue reperfusion is driven predominantly by microcirculatory spasm, microembolization, and excessive activation of coagulation factors, whereas, with increasing ischemia time, factors such as myocyte swelling, intramyocardial hemorrhage, and tissue necrosis become predominant.

A strong prognostic factor for the occurrence of reperfusion disorders after successful pPCI was the time from the onset of the first pain symptoms. The same dependence was observed in the time from the onset of symptoms to conducting pPCI. A similar correlation was found in a study conducted by Mazhar et al., which found that the time between the onset of chest pain and PCI > 360 min was the main prognostic factor for impaired reperfusion. NRP was also associated with higher mortality during the 12-month follow-up period. Prolonged ischemia time leads to progressive microcirculation damage—from endothelial edema to vessel obstruction—which significantly reduces the chances of effective tissue reperfusion despite angiographic success of the procedure [24]. Our results confirm this relationship. This emphasizes that reducing the total ischemic time remains the most important strategy for preventing NRP.

Another factor that may potentially increase the risk of no reflow is predilatation. This procedure may lead to fragmentation of the thrombus or unstable atherosclerotic plaque, which promotes their dislocation and microembolization within the coronary microcirculation. In the present study, predilatation performed before stent implantation was significantly more frequent in patients with impaired tissue reperfusion and was an independent risk factor for the no-reflow phenomenon. Similar observations were made in the PIHRATE study, in which the use of direct stent implantation preceded by manual aspiration thrombectomy was associated with better microcirculation perfusion, assessed by the MBG scale [25]. However, these results should be interpreted with caution, as patients with impaired tissue reperfusion often present with more advanced coronary atherosclerosis, as reflected by a significantly higher SYNTAX score, which, in many cases, precludes direct stenting. To better disentangle the effect of predilatation from underlying lesion complexity, future studies should incorporate detailed angiographic characterization of thrombus burden, plaque ulceration, and lesion length and diameter. In the study by Mahmoud Tantawy et al., it was also shown that the use of a larger diameter balloon during predilatation was associated with a significantly higher risk of no reflow compared to the use of smaller balloons (14.2% vs. 2.7%; *p* < 0.001) [26]. In the study by Shan-Shan Zhou et al., the combined use of high-dose atorvastatin, intracoronary adenosine, GP IIb/IIIa inhibitor, and aspiration thrombectomy in high-risk patients with no reflow was associated with a significant reduction in the incidence of no reflow compared to the standard treatment group (2.8% vs. 35.2%; *p* < 0.001) [27]. Such an association was not demonstrated in our study.

MI of the anterior wall, most commonly associated with obstruction of the left anterior descending artery (LAD), embraces the largest area of the left ventricular muscle. As a result, it affects a greater mass of ischemic and necrotic myocardium, which contributes to a higher risk of adverse outcomes. The study by Kandzari et al. showed that patients with anterior STEMI had a higher risk of death and heart failure compared to non-anterior STEMI patients [28]. In CMR-based analysis by Reindl et al., patients with anterior STEMI had greater extent of myocardial damage and a higher risk of MACE (HR 2.01; 95% CI 1.05–3.83; *p* = 0.03). However, after taking into account the extent of necrosis, it turned out that it was the size of the infarct—and not the localization itself—that was an independent predictor of worse prognosis (HR 1.03; 95% CI 1.01–1.06; *p* = 0.01). Also, patients with anterior STEMI demonstrated a non-significant trend towards greater microvascular obstruction (*p* = 0.09). It suggests that worse prognosis in LAD obstruction is primarily due to the larger mass of damaged myocardium and the occurrence of microcirculation disorders [29]. Our findings are consistent with prior studies; the no-reflow phenomenon was significantly more frequent in patients with anterior wall infarction, with anterior localization independently predicting impaired reperfusion. Combining the available data, it can be concluded that anterior localization of STEMI should be considered as a marker of particularly high risk, and the worse prognostic results are mainly due to the greater extent of myocardial damage related to the area of vascularization by the LAD. Higher Killip–Kimball class in acute phase of MI reflects more advanced heart failure and greater extent of myocardial damage. In our analysis it was one of the strongest independent predictors of no reflow (OR = 7.69; 95% CI 1.88–31.38; *p* = 0.004), which indicates that patients presenting signs of circulatory failure before the procedure are particularly vulnerable to reperfusion disorders at the microcirculation level. This relationship has a pathophysiological justification. A higher Killip class is associated with a larger mass of myocardium at risk, more severe inflammation, interstitial edema, and higher left ventricular filling pressure; all these factors contribute to microcirculation damage and intensify the phenomenon of microvascular obstruction, which is the key element of no-reflow phenomenon. The results of our study are consistent with the observations of other authors who indicated that patients in Killip class ≥2 have a significantly increased risk of impaired reperfusion after pPCI and a higher mortality in long-term follow-up [30,31].

The results of the analysis confirm that the occurrence of the no-reflow phenomenon after pPCI is an unfavorable prognostic factor. In the study population, patients with impaired reperfusion had significantly higher mortality both during hospitalization and during the 12-month follow-up. In a prospective study by Ndrepepa et al., the presence of this phenomenon was associated with more than twice the death rate in a 5-year follow-up (18.2% vs. 9.5%; OR: 2.02, 95% CI: 1.44 to 2.82; *p* < 0.001) [32]. Similar conclusions were drawn in the analysis by Kim et al., which showed that both transient and persistent no reflow after pPCI are associated with significantly worse clinical outcomes, including higher mortality and rehospitalization rates due to heart failure [33]. This phenomenon reflects persistent impairment of coronary microcirculation, which, despite effective angiographic revascularization of the infarct-related artery, leads to insufficient myocardial perfusion and larger area of myocardial necrosis. As a consequence, there is unfavorable left ventricular remodeling, progressive failure and a worse long-term prognosis. The results of our study confirm that patients with no-reflow require more intensive surveillance and optimization of therapy after intervention, including pharmacological strategies aimed at protecting the microcirculation. Considering that ischemia time is one of the strongest factors determining microcirculation damage and the development of no-reflow actions aimed at shortening it remain crucial, this requires both extensive public education to reduce patient delays and an efficiently functioning network of catheterization laboratories to minimize pain-to-door and pain-to-wire times, which are fundamental in preventing no reflow.

### Strengths and Limitations

The main strength of this study is its prospective design, which enabled consistent and systematic collection of clinical, angiographic, and laboratory data in a homogeneous STEMI population treated within a single high-volume PCI center. The use of an inclusive definition of impaired reperfusion, which combined angiographic and electrocardiographic parameters, allowed for an accurate assessment of microvascular dysfunction. Detailed evaluation of procedural characteristics with serial measurements of platelet reactivity, provided a comprehensive analysis of factors contributing to the no-reflow phenomenon. Additionally, the inclusion of 12-month mortality follow-up allowed for the assessment of the prognostic implications of no reflow.

However, this study also has important limitations. Its relatively small sample size and single-center observational design limit generalizability and reduce statistical power, especially in multivariate models. Notably, despite the application of relatively inclusive criteria, patient recruitment took longer than expected. This limitation should be considered when interpreting the stability and generalizability of the identified independent predictors. The lack of additional techniques for microcirculatory assessment, such as cardiac magnetic resonance or invasive microvascular measurements, may have led to under- or overestimation of microvascular obstruction. Moreover, platelet reactivity was assessed only twice, which may not fully capture dynamic changes in the early post-PCI period. Finally, apart from aspirin, resistance was assessed only for clopidogrel; therefore, generalization of our findings to the entire class of P2Y12 inhibitors should be approached with caution.

## 5. Conclusions

The main findings of our study demonstrated that the no-reflow phenomenon occurred frequently (39%) and translated into clearly worse short- and mid-term outcomes, with higher mortality both during hospitalization and at 12-month follow-up. Impaired reperfusion was more common in older patients and those with reduced kidney function, whereas traditional risk factors such as hypertension, diabetes, and hyperlipidemia did not show a significant association. Interestingly, no reflow was less prevalent among chronic smokers. The multivariate analysis identified four major independent predictors of impaired reperfusion: longer pain-to-balloon time, anterior wall myocardial infarction, the need for predilatation, and higher Killip–Kimball class at admission, all of which reflect a larger ischemic burden and more advanced hemodynamic compromise at presentation. In contrast to some earlier reports, platelet reactivity measured on admission and on the fifth day of hospitalization was not associated with no reflow, suggesting that, in this population, microcirculatory damage was driven predominantly by ischemic time, infarct extent, and procedural factors rather than by residual platelet hyperreactivity under standard antiplatelet therapy. Overall, these findings emphasize that reducing total ischemic time, avoiding unnecessary predilatation where feasible, and recognizing anterior STEMI patients with higher Killip class as a particularly vulnerable group should be key priorities in preventing the no-reflow phenomenon and improving prognosis after primary pPCI.

## Figures and Tables

**Table 1 diagnostics-16-00149-t001:** Baseline characteristics of the studies population.

	Variable	Normal Reperfusion (61)	Impaired Reperfusion (39)	*p*-Value
Baseline data	Age	62.8 (11.6)	70.6 (12.01)	0.002
	Male sex	45 (74%)	22 (56%)	0.07
	Smoking	35 (57.4%)	9 (23.1%)	<0.001
	Hypertension	36 (59%)	27 (69.2%)	0.3
	Diabetes	11 (18%)	9 (23.1%)	0.54
	Hyperlipidemia	48 (78.7%)	27 (69.2%)	0.29
	Atrial fibrillation	5 (8.2%)	8 (20.5%)	0.07
	eGFR (mL/min)	86 (72–114)	71 (51–92)	0.005
	Creatinine (mg/dL)	0.86 (0.81–0.96)	0.95 (0.82–1.18)	0.07
	Previous CAD	9 (14.8%)	10 (25.6%)	0.18
	Previous MI	5 (8.2%)	2 (5.1%)	0.56
	Previous revascularization	5 (8.2%)	1 (2.6%)	0.25
	EF (%)	41.3 (7.9)	34.3 (8.6)	<0.001
	Pain time (min)	163 (120–236)	466 (103–771)	<0.001
	Pain to drug (min)	90 (60–172)	360 (100–669)	<0.001
	Pain to balloon (min)	185 (147–271)	505 (238–815)	<0.001
	Duty hours	23 (37.7%)	10 (25.6%)	0.21
	Sudden cardiac arrest	3 (4.9%)	3 (7.7%)	0.57
MI Location	Anterior MI	25 (41%)	29 (74.4%)	0.004
	Inferior MI	33 (54.1%)	10 (25.6%)	
	Lateral MI	3 (4.9%)	0 (0%)	
Platelet reactivity	ADP test on admission	869 (609–1274)	804 (419–1150)	0.24
	ASPI test on admission	210.5 (82–487)	235 (138–379)	0.6
	ADP test on fifth day	266.5 (175.5–484)	349 (203–515)	0.18
	ASPI test on fifth day	127 (76–191)	139 (89–211)	0.67
	HTPR on admission (cut-off 416 AUC)	49 (80.3%)	25 (64.1%)	0.15
	HTPR of fifth day (cut-off 416 AUC)	14 (23%)	11 (28.2%)	0.59
	Clopidogrel resistance (>46 AU)	27 (44.3%)	23 (59%)	0.15
Drugs on discharge	Clopidogrel	61 (100%)	38 (97.4%)	1
	ASA	61 (100%)	38 (97.4%)	1
	ACEi	60 (98.4%)	32 (82.1%)	0.008
	B-blocker	59 (96.7%)	36 (92.3%)	0.63
	Statin	59 (96.7%)	36 (92.3%)	0.63
	MRA	24 (39.3%)	25 (64.1%)	0.01
	Diuretics	22 (36.1%)	28 (71.8%)	<0.001
	PPI	59 (96.7%)	37 (94.9%)	0.86
Follow-up	Death in 12 months	2 (3.3%)	10 (25.6%)	0.001
	Death during hospitalization	1 (1.6%)	4 (10.3%)	0.054

Abbreviations: ADP test, adenosine diphosphate induced platelet aggregation test; ASA, acetylsalicylic acid; ASPI test, arachidonic acid induced platelet aggregation test; CAD, coronary artery disease; EF, left ventricular ejection fraction; eGFR, estimated glomerular filtration rate; HTPR, high on-treatment platelet reactivity; MRA, mineralocorticoid receptor antagonist; PPI, proton pump inhibitor.

**Table 2 diagnostics-16-00149-t002:** Angiographic data.

Variable		Normal Reperfusion (61)	Impaired Reperfusion (39)	*p*-Value
SYNTAX Score I		17 (8)	21 (9)	0.04
TIMI flow before pPCI	0	40 (65.6%)	31 (79.5%)	0.46
	1	2 (3.3%)	1 (2.6%)	
	2	11 (18%)	5 (12.8%)	
	3	8 (13.1%)	2 (5.1%)	
Thrombectomy		31 (50.1%)	23 (60%)	0.43
Predilation		31 (50.1%)	34 (87.2%)	<0.001
Number of stents	0	0 (0%)	1 (2.6%)	0.43
	1	46 (75%)	32 (82.1%)	
	2	12 (20%)	4 (10.3%)	
	3	2 (3.3%)	2 (5.1%)	
	4	1 (1.6%)	0 (0%)	
Stent data	Mean stent diameter	3.5 (3–3.5)	3.5 (3–3.5)	0.51
	Mean stent length	22 (16–28)	22 (18–28)	0.99
TIMI flow after pPCI	0	0 (0%)	0 (0%)	<0.001
	1	0 (0%)	3 (7.7%)	
	2	0 (0%)	22 (56.4%)	
	3	61 (100%)	14 (35.9%)	
MBG after pPCI	0	0 (0%)	1 (2.6%)	<0.001
	1	0 (0%)	7 (17.9%)	
	2	10 (16.4%)	18 (46.2%)	
	3	51 (83.6%)	13 (33.3%)	
TMPG after pPCI	0	0 (0%)	3 (7.7%)	<0.001
	1	3 (4.9%)	6 (15.4%)	
	2	11 (18%)	19 (48.7%)	
	3	47 (77%)	11 (28.2%)	
cTFC after pPCI		20 (15–27)	43 (24–58)	<0.001
Time of procedure		32 (25–48)	40 (30–55)	0.07
X-ray time (min)		7.6 (5.1–10.5)	10.3 (6.9–15.7)	0.02
Radiation dose (mGy)		1211 (775–1639)	1340 (834–2138)	0.27
Contrast dye (mL)		150 (120–170)	170 (140–220)	0.07
Use of GP IIb/IIIa inhibitors		23 (37.7%)	18 (46.2%)	0.4
No early post-pPCI ST resolution (<50% reduction of elevation)		0 (0%)	29 (74.4%)	<0.001

Abbreviations: cTFC, corrected TIMI frame count; GP IIb/IIIa, glycoprotein IIb/IIIa inhibitor; MBG, myocardial blush grade; mGy, milligray; pPCI, primary percutaneous coronary intervention; TIMI, thrombolysis in myocardial infarction flow grade; TMPG, TIMI myocardial perfusion grade.

**Table 3 diagnostics-16-00149-t003:** Multivariate logistic regression. Predictors of impaired reperfusion.

Variable	OR (95% CI)	*p*-Value
Male sex	0.77 (0.2–3.01)	0.71
Pain to balloon (per increase in 10 min)	1.05 (1.02–1.07)	<0.001
Anterior MI	5.05 (1.14–22.38)	0.03
Predilation	7.66 (1.78–32.9)	0.006
SYNTAX score I (per increase in 1 point)	0.94 (0.87–1.02)	0.13
Killip–Kimball Class (per increase in 1 class)	7.69 (1.88–31.38)	0.004
Smoking	0.3 (0.08–1.1)	0.07
Atrial fibrillation	0.37 (0.05–2.87)	0.34

Abbreviations: CI, confidence interval; MI, myocardial infarction; OR, odds ratio; SYNTAX score I, coronary lesion complexity score.

## Data Availability

The data presented in this study are available on request from the corresponding author due to reasonable request.

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
