# Peer review of "Predictors of Impaired Reperfusion in ST-Elevation Myocardial Infarction Treated with Primary PCI: Preliminary Results from COMA.NET Project"

_diagnostics, 2026, doi:10.3390/diagnostics16010149_

Round 1
Reviewer 1 Report
Comments and Suggestions for Authors
Abstract: is in structural form generally well written.
Keywords: OK.
Introduction: well written, explained in logical order in detail, addressing to “what we know” so far and what is unclear about no reflow phenomenon.
Materials and Methods
Chapter is well written, study design, study period, inclusion and exclusion criteria, procedure with participant and statistical methods ate very good described, it is possible to repeat study according to description. I have two quwstions:
- Why 24 h is used as maximum time of symptom onset and not 48h ?
- Why patient preloaded with prasugrel or ticargelor were excludes (ESC guidelines prefer listed over clopidogrel which was allowed) ?
Results are well presented in understandable way to reader including tables and figures.
Discussion is excellent, authors addressed to theirs results point by point with relevant references pointing concordance and differences as well as study limitations.
Conclusion: is well written with clear “take home messages” – educational especially about lesion predilataion.
Literature: well written, no instructions.
Author Response
Reviewer 1.
Abstract: is in structural form generally well written.
Keywords: OK.
Introduction: well written, explained in logical order in detail, addressing to “what we know” so far and what is unclear about no reflow phenomenon.
Materials and Methods
Chapter is well written, study design, study period, inclusion and exclusion criteria, procedure with participant and statistical methods ate very good described, it is possible to repeat study according to description. I have two quwstions:
1. Why 24 h is used as maximum time of symptom onset and not 48h ?
Reply 1: Thank you for this comment. As the Reviewer kindly mentioned, current guidelines recommend pPCI in cases with symptom onset less than 48h. The preferred time window for primary revascularization with a class I level of recommendation is within 12 hours from the onset of symptoms in STEMI, whereas for the 12–48-hour window the strength of recommendation is class IIa. These recommendations are present both in the 2017 ESC guidelines and in the current 2023 ESC guidelines. However, the main reason for limiting the duration of anginal symptoms to 24 hours in our study was to balance the underlying mechanisms contributing to the no-reflow phenomenon. As described in the literature, during the early phase of myocardial infarction, impaired tissue reperfusion is driven predominantly by microembolization and excessive activation of coagulation factors, whereas with increasing ischemia time, factors such as myocyte swelling, microvascular hemorrhage, and necrosis become dominant. Therefore, during study design, the maximum time from symptom onset was restricted to 24 hours.
Changes 1: None.
2. Why patient preloaded with prasugrel or ticargelor were excludes (ESC guidelines prefer listed over clopidogrel which was allowed) ?
Reply 2: Thank you for this excellent question. The ESC guidelines recommending the use of more potent P2Y12 receptor inhibitors over clopidogrel in STEMI were published in 2017, during the enrollment period of the study. In Poland, widespread implementation of these newer antiplatelet agents in routine clinical practice for the majority of patients began around 2018. To ensure a uniform study population and, in particular, to avoid the potential confounding effect of altered platelet inhibition associated with more potent antiplatelet therapy, individual patients who received prasugrel or ticagrelor were excluded from the study.
Changes 2: None.
Results are well presented in understandable way to reader including tables and figures.
Discussion is excellent, authors addressed to theirs results point by point with relevant references pointing concordance and differences as well as study limitations.
Conclusion: is well written with clear “take home messages” – educational especially about lesion predilataion.
Literature: well written, no instructions.
Reply:
Thank you for time invested in reviewing our manuscript and valuable comments.
Reviewer 2 Report
Comments and Suggestions for Authors
This paper examines the no-reflow phenomenon, an important and relevant issue in reperfusion treatment for myocardial infarction.
The following are significant caveats regarding the content of the paper:
Regarding the "Materials and Methods" section:
1) The analysis is stated to be prospective, yet data from patients from 2015-2017 are presented. A clarification is needed (optional, not included in the article text) regarding why data from 10 years ago is being presented.
2) Only patients receiving clopidogrel were included, with exclusion criteria including prasugrel and ticagrelor (despite the patients' sinus rhythm?). This circumstance significantly limits the value of the paper. Clopidogrel is being used less frequently in line with current trends. Consequently, the value of the analysis using this drug is significantly reduced. This limitation is quite significant and calls into question the clinical utility of the paper in 2025.
3) The sample size is also concerning. If all consecutive patients admitted from August 2015 to August 2017 (i.e., over a two-year period) were included, then why were so few patients—only 100—meeting the fairly standard inclusion criteria? Clarification is needed.
Regarding the "Results" section:
1) What was the basis for selecting the no-reflow criteria? Was this the authors' own model? If not, a referent to the original source of these criteria is required. I must note that I fully agree with the authors' chosen criteria, but I am unsure of their universal applicability. It would have been better if the authors had clearly articulated the no-reflow criteria in the "Materials and Methods."
2) The authors state that no-reflow was associated with a lower left ventricular ejection fraction. However, logically, echocardiography was assessed after myocardial revascularization, and therefore this parameter cannot be considered "baseline data." In my opinion, a decrease in LVEF should be considered a consequence of no-reflow and suboptimal myocardial revascularization. If I have misinterpreted the information in the text, please correct me and indicate the time of the echocardiogram.
Author Response
Reviewer 2.
This paper examines the no-reflow phenomenon, an important and relevant issue in reperfusion treatment for myocardial infarction.
The following are significant caveats regarding the content of the paper:
Regarding the "Materials and Methods" section:
- The analysis is stated to be prospective, yet data from patients from 2015-2017 are presented. A clarification is needed (optional, not included in the article text) regarding why data from 10 years ago is being presented.
Reply 1. Thank you for this comment. Indeed, the protocol for this prospective study was established a decade ago. Data collection spanned a 2-year period, followed by comprehensive analysis and a minimum of 1 year of follow-up observation. The core objective of the study was to investigate factors contributing to impaired myocardial tissue reperfusion. A broad array of clinical variables, platelet function metrics, and biochemical markers were evaluated. Previous conference abstracts and publications from our institution had explored correlations between select biochemical markers, such as GDF-15, and the no-reflow phenomenon. These data also formed the foundation for a doctoral dissertation. The manuscript under consideration elucidates the impact of clinical factors and platelet reactivity on impaired tissue reperfusion. These specific data have not been previously published from our database.
Changes 1. None
- Only patients receiving clopidogrel were included, with exclusion criteria including prasugrel and ticagrelor (despite the patients' sinus rhythm?). This circumstance significantly limits the value of the paper. Clopidogrel is being used less frequently in line with current trends. Consequently, the value of the analysis using this drug is significantly reduced. This limitation is quite significant and calls into question the clinical utility of the paper in 2025.
Reply 2. Thank you very much for this comment. We fully agree with the Reivewer that both ticagrelor and especially prasugrel are the preferred P2Y12 inhibitors in majority patients with STEMI. The ESC guidelines recommending the use of more potent P2Y12 receptor inhibitors over clopidogrel in STEMI were published in 2017, during the enrollment period of the study. In Poland, widespread implementation of these newer antiplatelet agents in routine clinical practice for the majority of patients began around 2018. To ensure a uniform study population and, in particular, to avoid the potential confounding effect of altered platelet inhibition associated with more potent antiplatelet therapy, individual patients who received prasugrel or ticagrelor were excluded from the study. Please note that the actual aim of the study was to assess factors potentially influencing no-reflow phenomenon and platelet reactivity was only one of the variables included in final analysis. Finally, both uni- and multivariate analyses did not demonstrate that platelet reactivity (analyzed using both continuous variables and cutoff values) to aspirin and clopidogrel influences the occurrence of the no-reflow phenomenon. Therefore, in our opinion, the topic is very relevant even in 2025/2026 as it shows that platelet reactivity and resistance to clopidogrel and aspirin does not play a crucial role in no-reflow phenomenon occurrence. Please note that results from other studies are inconclusive. Even early administration of glycoprotein IIb/IIIa inhibitors, reduces microvascular obstruction (MVO) but does not yield a net clinical benefit when considering the increased incidence of bleeding (REVERSE-FLOW trial). Data from the meta-analysis by Yike Li et al. (2023) indicate that ticagrelor, compared to clopidogrel, reduces microvascular dysfunction (assessed by various methods); however, it does not provide insight into the impact on long-term clinical outcomes. In summary, although the findings from our study cannot be directly extrapolated to patients receiving more potent antiplatelet therapy, current literature has not demonstrated an independent effect of stronger platelet inhibition on long-term prognosis with respect to reducing the no-reflow phenomenon (NRP). Based on our data, we believe that other factors exert a stronger influence on the incidence of NRP, particularly in patients with a prolonged pain‑to‑balloon time, as observed in our cohort. Accordingly, we believe that this limitation does not undermine the scientific or clinical value of the present study.
Changes 2. Please see discussion and limitations section amended.
- The sample size is also concerning. If all consecutive patients admitted from August 2015 to August 2017 (i.e., over a two-year period) were included, then why were so few patients—only 100—meeting the fairly standard inclusion criteria? Clarification is needed.
Reply 3. Thank you for this question, we are glad to clarify. We included 100 consecutive patients with STEMI who received clopidogrel as a P2Y12 inhibitor. In our centre, there are around 250-300 MI cases yearly, one-third of them are STEMI cases.
The slowdown in patient recruitment, particularly during the second year, was primarily due to the change in the ESC guidelines in 2017, which introduced the use of more potent antiplatelet agents instead of clopidogrel. This partially reduced the number of patients who had not received a loading dose of these agents prior to enrollment. It should also be noted that, according to the exclusion criteria, patients in cardiogenic shock and those with atrioventricular or intraventricular conduction abnormalities on baseline ECG were not enrolled, further limiting the number of eligible candidates.
Changes 3. Please see limitations section revised.
Regarding the "Results" section:
- What was the basis for selecting the no-reflow criteria? Was this the authors' own model? If not, a referent to the original source of these criteria is required. I must note that I fully agree with the authors' chosen criteria, but I am unsure of their universal applicability. It would have been better if the authors had clearly articulated the no-reflow criteria in the "Materials and Methods."
Reply 1. We would like to thank the Reviewer for this valuable comment. When designing the methodology of this study and defining the criteria for impaired tissue reperfusion, our aim was to identify a simple and practical approach capable of assessing both early and late mechanisms of microvascular reperfusion injury following myocardial infarction. Based on existing literature, we decided to combine an angiographic scale with an electrocardiographic one. Similar combinations of assessment scales have been proposed in previously published studies [1,2].
It is well established that impaired tissue reperfusion after restoration of flow in the infarct-related artery (IRA) during STEMI has a significant impact on patients’ long-term prognosis. In selecting the angiographic assessment, we analyzed angiograms using the TIMI, MBG, TMPG, and cTFC scales. Each of these parameters was subsequently evaluated in univariate logistic regression to determine its predictive value for 1-year mortality. Among the electrocardiographic parameters, ST-segment resolution (STR) at 60 minutes after pPCI—with thresholds of 50% and 70%—was analyzed in the same way. TIMI flow grade <3 and STR <50% demonstrated favorable predictive performance, and their combination also showed good prognostic value in predicting long-term mortality. Therefore, the combined use of these two scales was selected for the analysis. The corresponding source data are presented in the tables.
|
|
Odds ratio (OR) |
CI +/-95% |
p |
|
TIMI <3 |
2,890 |
1,500 - 5,569 |
0,002 |
|
MBG <3 |
1,688 |
0,912 - 3,123 |
0,096 |
|
TMPG<3 |
3,176 |
0,88 - 11,36 |
0,076 |
|
cTFC >27 frames |
2,514 |
1,260 - 5,017 |
0,009 |
|
|
Odds ratio (OR) |
CI +/-95% |
p |
|
STR < 50% |
2,488 |
1,301 - 4,759 |
0,006 |
|
STR < 70% |
1,505 |
0,757 - 2,993 |
0,244 |
|
|
Odds ratio (OR) |
CI +/-95% |
p |
|
STR < 50% and/or TIMI <3 |
3,189 |
1,446 - 7,034 |
0,004 |
- Sorajja P., Gersh B.J., Costantini C. Combined prognostic utility of ST-segment recovery and myocardial blush after primary percutaneous coronary intervention in acute myocardial infarction. Eur Heart J. 2005;26:667–674. doi: 10.1093/eurheartj/ehi167.
- Giugliano R.P., Sabatine M.S., Gibson C.M. Combined assessment of thrombolysis in myocardial infarction Flow grade, myocardial perfusion grade and ST-segment resolution to evaluate epicardial and myocardial reperfusion. Am J Cardiol. 2004;93:1362–1367. doi: 10.1016/j.amjcard.2004.02.031.
- Changes 1. Please see Material and Methods section revised.
- The authors state that no-reflow was associated with a lower left ventricular ejection fraction. However, logically, echocardiography was assessed after myocardial revascularization, and therefore this parameter cannot be considered "baseline data." In my opinion, a decrease in LVEF should be considered a consequence of no-reflow and suboptimal myocardial revascularization. If I have misinterpreted the information in the text, please correct me and indicate the time of the echocardiogram.
Reply 2. Thank you for this accurate comment. Patients with STEMI undergo primary PCI as soon as possible after hospital admission. In selected cases, when mechanical complications of myocardial infarction or ST-segment elevation due to other causes are suspected, bedside transthoracic echocardiography is performed prior to the procedure. However, in the majority of cases, TTE in STEMI patients is performed after pPCI or on the following day. We agree with the Reviewer that reduced LVEF is a consequence of the extent and duration of ischemia as well as impaired tissue reperfusion, rather than a causative factor leading to the no-reflow phenomenon. Such a hypothesis could only be verified through retrospective analysis of TTE recordings performed before hospital admission for STEMI. The corresponding revisions have been incorporated into the manuscript.
Changes 2. Please see Abstract, Results and Discussion section. Please see multiple minor changes across manuscript.
Reply:
Thank you for time invested in reviewing our manuscript and valuable comments.
Please also find the reply to the reviewers’ comments in the attached file.

Round 2
Reviewer 2 Report
Comments and Suggestions for Authors
After the changes and clarifications from the authors, the article took on a better appearance